# Anticoagulants: A Short History, Their Mechanism of Action, Pharmacology, and Indications

**DOI:** 10.3390/cells11203214

**Published:** 2022-10-13

**Authors:** Marco Heestermans, Géraldine Poenou, Hind Hamzeh-Cognasse, Fabrice Cognasse, Laurent Bertoletti

**Affiliations:** 1French Blood Establishment (EFS) Auvergne Rhône Alpes-Scientific Department, F-42270 Saint-Etienne, France; 2INSERM, U1059, SAINBIOSE, Jean Monnet University, F-42023 Saint-Etienne, France; 3Service de Médecine Vasculaire et Thérapeutique, CHU de St-Etienne, F-42055 Saint-Etienne, France; 4INSERM, CIC-1408, CHU Saint-Etienne, F-42055 Saint-Etienne, France

**Keywords:** venous thromboembolism, anticoagulant drugs, heparin, vitamin K antagonists, DOACs, pharmacology, drug–drug interactions, novel anticoagulants

## Abstract

Anticoagulant drugs antagonize coagulation and are used to prevent or cure (recurrent) venous thromboembolism (VTE). Drugs to prevent clotting have been used for more than a century, and, nowadays, physicians possess a broad panel of multiple anticoagulants to meet the individual needs of a patient. Within this review, we aimed to revise the history of the different anticoagulants that are currently prescribed in the clinic. In addition, we compared their pharmacological properties, medical indications, and the difficulties in implementing new anticoagulants in vulnerable patient populations. Since the introduction of unfractionated heparin in the 1930s, major advances in the mechanistic understanding and the medical use of anticoagulants have allowed for significant improvements to treat VTE patients. However, a new generation of anticoagulants is currently being tested in clinical trials, with the goal of further optimizing medical care.

## 1. Introduction

Thrombosis is the leading cause of death worldwide, and, in terms of prevalence, the first among the non-contagious diseases for which the WHO aims to reduce the incidence (https://www.who.int/news-room/fact-sheets/detail/noncommunicable-diseases (accessed on 1 September 2022)). The disease is a result of vascular occlusion and, in clinics, is mainly manifested by myocardial infarction (5 per 1000 individuals per year) [1], stroke (1.3–4.1 per 1000 individuals per year) [2], or venous thromboembolism (VTE, 1–3 per 1000 individuals per year) [3]. Antiplatelet drugs are mostly used to treat arterial thrombotic events such as myocardial infarction and stroke, while VTE is prevented and cured using anticoagulants. As the name already implies, anticoagulants antagonize coagulation, i.e., secondary hemostasis [4]. Hemostasis is the physiological process where, in healthy individuals, vascular disruptions and bleedings as a result of an external aggression are stopped via the formation of a blood clot. In a pathological setting, coagulation drives the formation of undesired intravascular blood clots, which are referred to as thrombi [5]. 

The first anticoagulant drugs to prevent VTE, unfractionated heparin and warfarin, were identified by serendipity in the early 20th century [6,7]. In the following years, these drugs have been optimized to prevent or treat VTE more efficiently. More recently, a novel generation of anticoagulant drugs has been introduced, which are designed to target coagulation factor IIa (thrombin) and coagulation factor Xa (FXa), two enzymes that are crucial for coagulation [8,9]. Depending on the medical indication, nowadays, physicians can prescribe different anticoagulant drugs that are best-suited for the needs of their patients. However, although anticoagulant drugs are effective in preventing VTE, they still have constraints in terms of maneuverability and associated hemorrhagic risk [10]. The goal of this review is to revise the history and pharmacology of the different anticoagulants that are currently used in the clinic, and to briefly outline the challenges that face the community of thrombosis and hemostasis to improve our current anticoagulants. 

## 2. A Short History of Anticoagulants: From Serendipity to Molecular Design

### 2.1. Unfractionated Heparin

Unfractionated heparin (UFH) was the first anticoagulant drug, discovered in 1916 by a medical student at Johns Hopkins University, Jay McLean, while studying the putative prothrombotic properties of dog liver and heart extracts [11]. Surprisingly, the compound, later named heparin (since it was identified in the dog’s liver), demonstrated remarkably strong anticoagulant effects [12]. In the 1930s, several labs were able to purify UFH, and the compound was approved for clinical use. For the next couple of decades, UFH was used to prevent clotting in humans, without a real understanding of the mechanism of action [13]. It was not until the 1970s that it was discovered that UFH is a mixture of polymorphic polysaccharide chains obtained after the purification of vertebrate organs. These polysaccharide chains significantly enhance the activity of the major natural inhibitor of coagulation, antithrombin, which in healthy individuals accounts for the majority of all natural anticoagulation [14]. Antithrombin is a serine protease inhibitor that primarily antagonizes thrombin and FXa, two pivotal actors within the coagulation cascade [15] (Figure 1). Once antithrombin binds to thrombin or FXa, they form a complex that is rapidly degraded by the circulation [16]. The interaction of UFH with antithrombin is mostly mediated by a unique pentasaccharide sequence that is randomly distributed along the polysaccharide chains [17]. Since UFH accelerates the mode of action of antithrombin by approximately 1000 times, it functions as a highly efficient anticoagulant drug [18]. 

### 2.2. Vitamin K Antagonists

In the 1930s, biochemist Karl Paul Link was contacted to investigate the mysterious massacre of cattle on several farms surrounding his laboratory at the University of Wisconsin. It took him and his team several years to determine that the hay fed to the cattle was spoiled after a series of wet summers and that one of the main constituents of spoiled hay, sweet clover, contained a toxic component [19]. In 1939, dicumarol was extracted, and this turned out to be an effective vitamin K antagonist (VKA). In the bloodstream, dicumarol competes with vitamin K epoxide reductase, an enzyme that recycles vitamin K [20,21]. As a result, VKAs deplete circulating vitamin K over time. Several coagulation factors depend on vitamin K for their conversion from inactive zymogens to active enzymes [22]. In contrast to heparins, VKAs are suitable for oral admission, thus avoiding the mandatory injections required for heparins. Initially, dicumarol was used to prevent thrombosis, and from the 1960s onwards its derivative warfarin, named after the Wisconsin Alumni Research Foundation, has become the reference treatment for thromboembolic events [7]. 

### 2.3. Low-Molecular-Weight Heparin

In the 1960s–1970s, worldwide interest in antithrombotic research increased because of a growing awareness of the significant contribution of thrombosis to diseases such as stroke, myocardial infarction, and VTE [23]. During this period, the actors within the coagulation cascade were identified one by one, and this resulted in a better mechanistic understanding of the sequence of events leading to thrombotic diseases. This improved understanding, together with advances in biochemical techniques that allowed for major advances in the development of anticoagulants. Different research groups within Europe and the US started working on the development of a subclass of heparins that are based on organ-derived UFH, the so-called low-molecular-weight heparins (LMWHs) [24]. LMWHs are derived from UFH by chemical splitting, into approximately one-third of the original size. The principal mechanism of action is similar for UFH and LMWHs (Figure 1), but LMWHs have fewer side effects and produce a more predictable anticoagulant response [25]. Shortly after the LMWH preparations were standardized, the drugs were tested in clinical trials, and LMWHs such as enoxaparin have now replaced heparin for most thrombotic indications [26,27]. Since LMWHs are injected subcutaneously in a fixed dose without the need for consistent anticoagulant monitoring, these drugs paved the way for treating some patients with VTE as outpatients, rather than at the hospital [28,29].

### 2.4. Fondaparinux, Argatroban, and Bivalirudin

In recent years, rapid advances in biotechnology have enabled “designing” molecules or drugs with a predetermined purpose [30]. To develop new anticoagulants, molecules can be generated that antagonize specific clotting factors. Structural chemists now fabricate small molecules designed to fit into the active component of clotting enzymes, like a key into a lock [31]. Due to this specificity, these novel drugs normally coincide with fewer off-target effects and drug–drug interactions. Fondaparinux is a small molecule with a structure that is based on the active component of heparins, and it was the first of a new class of selective antithrombin-dependent FXa inhibitors (Figure 1) [32]. Compared to UFH and LMWH, although it is more expensive to fabricate the much smaller synthetic molecule, fondaparinux has more predictable pharmacokinetics that render intensive monitoring and dose adjustment unnecessary. Argatroban is a drug derived from L-arginine and is able to specifically block the active site of thrombin, thus functioning independently of antithrombin (Figure 1) [33]. 

For the development of novel anticoagulants, inspiration can be found from anticoagulant toxins produced by a variety of animals. Anti-clotting proteins from an array of exotic creatures, such as ticks, leeches, snakes, and vampire bats, can be studied, extracted or produced in vitro, genetically engineered, and used as drugs upon mass chemical synthesis [34]. A synthetic drug developed using this approach is bivalirudin. Bivalirudin is an anticoagulant molecule based on the structure of hirudin, an anti-clotting substance found in leeches [35,36]. Bivalirudin is a direct thrombin inhibitor and was introduced as an alternative to heparins, to treat patients suffering from heparin-induced thrombocytopenia [37] (Figure 1). Although fondaparinux, argatroban, and bivalirudin present significant benefits over heparins because of their pharmacokinetics, they continue to rely on parenteral admission. 

### 2.5. Direct Oral Anticoagulants

The biggest advantage of VKAs over heparins is their oral admission, even though the anticoagulant effect can be delayed for days, so intense monitoring is required. Therefore, until recently, patients requiring anticoagulants were, upon admission to a hospital, first given heparins for five to seven days, the period after which the anticoagulant effect of VKAs sets in. During this bridging period, the bleeding risk is also substantially increased, since two anticoagulant drugs are present at the same time in the circulation of the patient [38]. The newest generation of direct oral anticoagulants (DOACs) circumvents these issues because they are administered orally and have a rapid mode of action [39,40]. Since their introduction in 2010, DOACs have largely replaced VKAs in the clinic, as they do not require the heparin bridging period. DOACs are small molecules that occupy the catalytic site of either FXa or thrombin, preventing their capacity to cleave and activate their substrates [41] (Figure 1). Their mode of action allows for a wider therapeutic window, enables more lenient monitoring, and has a lower risk of drug–drug interaction. 

The direct thrombin inhibitor ximelagatran was the first DOAC that was generated based on computer modeling [42,43,44]. Unfortunately, in the majority of the clinical trials where ximelagatran was tested, the treatment coincided with asymptomatic elevations of liver transaminases levels through unknown mechanisms [45]. For this reason, ximelagatran was withdrawn from the market, which highlights one of the main issues with DOACs: their liver toxicity, because of a high distribution volume. Another direct thrombin inhibitor, dabigatran, was selected from a panel of chemicals with a structure similar to benzamidine-based α-NAPAP (N-alpha-(2-naphthylsulfonylglycyl)-4-amidinophenylalanine piperidide), a compound, for which it has been known since the 1980s, that serves as a powerful inhibitor of various serine proteases, including thrombin [46]. The addition of ethyl-ester and hexyloxycarbonyl carbamide hydrophobic side chains to α-NAPAP led to the orally absorbed prodrug dabigatran, which is used in clinics today [47]. In addition to directly blocking thrombin using dabigatran, the serine protease upstream of thrombin in the coagulation cascade, FXa, appeared to be an intriguing target for DOACs [48,49]. FXa was already identified as a promising target for the development of new anticoagulants in the early 1980s, but the feasibility of FXa inhibition to impede thrombosis formation was not tested before the end of that decade. In 1987, the first FXa inhibitor, the naturally occurring compound antistasin, was isolated from the salivary glands of the Mexican leech, *Haementeria officinalis* [50]. This specific compound did not pass clinical trials, but, in the 2000s, other FXa-antagonizing DOACs such as rivaroxaban, apixaban, and edoxaban have been introduced in clinics [51]. Nowadays, FXa DOACs are widely used and are slowly but surely taking over the role of VKAs and heparins as the primary drugs to be prescribed to VTE patients [52].

## 3. Pharmacokinetics 

### 3.1. UFH, LMWHs, and Fondaparinux 

UFH and its derivates, LMWHs and fondaparinux, are anticoagulants that are administered parenterally, either subcutaneously or intravenously [53]. The average molecular weight of UFH is around 15,000 Da, while LMWHs are approximatively three times smaller (5000 Da) [54]. The synthetically produced drug fondaparinux, which is based on the active component of heparins, is 1780 Da. Heparins promote the activity of antithrombin, and activity can be rehearsed by the sulfate protamine that binds to circulating heparins, forming a stable salt and diminishing its anticoagulant potential [55]. 

UFH is directly isolated from vertebrate tissues, and it has a number of limitations due to its unpredictable nature [56,57,58,59]. In the circulation, infused UFH has a half-life of 1–1.5 h and has peak activity 4–6 h after infusion. Since the introduction of LMWHs, drugs such as enoxaparin have largely replaced UFH in the clinic because of the lower risk of bleeding, decreased binding to plasma and endothelial proteins, and improved bioavailability [60]. The biggest disadvantage of UFH and LMWHs is that approximately 0.2% of treated individuals develop potentially life-threatening heparin-induced thrombocytopenia (HIT) [61,62]. In short, for an unknown reason, a small group of heparin-treated individuals can generate an immune response against heparin that is in complex with circulating platelet factor 4 (PF4) [63]. Heparin–PF4–IgG-immune complexes subsequently activate platelets via their FcγIIa receptors, causing thrombocytopenia and thrombosis. The synthetic drug fondaparinux has similar pharmacological characteristics as LMWHs but has less affinity with PF4, thus rarely inducing HIT, which makes it an appealing alternative treatment option [64]. The pharmacokinetic proprieties of UFH, the LMWH enoxaparin, and fondaparinux are summarized in Table 1.

### 3.2. Vitamin K Antagonists

VKAs were the first generation of anticoagulants that are administered orally, a significant advantage over the parenterally administered heparins. Until recently, VKAs were the standard of care for oral anticoagulant treatment. Vitamin K promotes the biosynthesis of γ-carboxyglutamic acid residues in vitamin K-dependent coagulation factors (FII, FVII, FIX, and FX) that is essential for biological activity [65]. VKAs inhibit VKCOR1, an enzyme responsible for reducing its cofactor vitamin K to its active form. As a result of VKA treatment, the pool of vitamin K in the circulation is gradually reduced and consequently coagulation is abrogated [20]. VKA activity can be rehearsed by injecting exogenous vitamin K that serves as a neutralizing agent [66].

Although oral admission is a great improvement, compared to parentally administered heparins, VKAs present three major flaws. First of all, VKA treatment only reaches its optimal therapeutic action 3 to 7 days after the first admission because it transiently depletes the circulating vitamin K pool. For this reason, a bridging period usually consisting of a fast-acting heparin is required [38,67,68]. Second, the dosage and admission of VKA must be monitored frequently with a coagulation test, where the international normalized ratio (INR) is determined, based on the results of a prothrombin time (PT) test performed in a laboratory [69]. Based on the patient’s INR in response to the treatment, the dosage of VKA is adjusted. Finally, numerous interactions between VKAs and other drugs or food that can influence the efficacy of the treatment have been described [70,71]. The pharmacokinetic proprieties of warfarin, the first described and most commonly used VKA worldwide, will serve as a reference for the other VKAs, which are summarized in Table 2.

### 3.3. Bivalirudin and Argatroban

Bivalirudin and argatroban are two small synthetic molecules that are administered parentally. They both serve as direct inhibitors of thrombin and were introduced as an alternative to heparins in patients that developed HIT [72,73]. Compared to UFH and LMWH, the risk for bleeding events is comparable, but bivalirudin possesses a lower immunogenic profile and exclusively binds to thrombin in the circulation [74]. It already reaches its peak activity several minutes after injection, and it is eliminated via the kidney or as a result of proteolytic cleavage [75,76]. In terms of general pharmacological characteristics, argatroban is comparable to bivalirudin, except for its prolonged time to peak (hours instead of minutes) and half-life (3–5 h instead of 1–1.5 h for bivalirudin). Moreover, clearance mostly takes place by hepatic hydroxylation, which has a direct effect on the endothelium, since argatroban is capable to upregulate the generation of nitric oxide [77]. This means that argatroban treatment could have additional beneficial antithrombotic side effects because of NO-mediated inhibition of platelet activation [78,79]. The pharmacokinetic proprieties of bivalirudin and argatroban are summarized in Table 3.

### 3.4. DOACs

Since their introduction during the second decade of the 21st century, DOACs are gradually replacing other anticoagulants to treat VTE patients. Currently four DOACs are prescribed in the clinic: dabigatran, which targets thrombin, and rivaroxaban, apixaban, and edoxaban, which target FXa [39,80]. The drugs have a similar mechanism of action, as they all block the active site of their respective target enzyme that is exposed upon activation [81]. DOACs are small synthetic molecules that are administered orally. They are rapidly absorbed in the bloodstream and, because of their direct inhibition of coagulation proteins, they form a significant improvement for daily clinical practice compared to VKAs. In addition, DOACs have significantly less contraindications with food and other drugs, which makes them more predictable, thus requiring less intense monitoring [8]. The neutralizing agent for dabigatran is a humanized Fab fragment of an antibody (idaruzicumab) that directly binds dabigatran and neutralizes its activity [82]. Andexanet alpha is a recombinant protein that can serve as a decoy for FXa DOACs (although, for edoxaban, the efficacy needs to be investigated), by mimicking the FXa active site without being able to engage in the formation of the prothrombinase complex [83]. Since andexanet alpha is an expensive product, prothrombin complex concentrates (PCCs) can alternatively be provided as a neutralizing agent to patients treated with rivaroxaban, apixaban, or edoxaban [84]. PPCs circumvent FXa DOACs, since they function downstream of FXa in the coagulation cascade. DOACs have substantially facilitated VTE treatment, both for patients and physicians, and in the coming decades DOACs will most likely be more and more prescribed instead of other anticoagulants. The pharmacokinetic proprieties of dabigatran, rivaroxaban, apixaban, and edoxaban are summarized in Table 4.

## 4. Medical Indications 

VTE patients treated with parentally administered drugs, i.e., heparin (derivates), bivalirudin, and argatroban, can experience major discomfort with imperative intravenous or subcutaneous injections. Oral anticoagulants are easier to administer, so, for this reason, they can be preferred to treat thrombotic diseases, depending on the individual patient’s needs and their medical indication [39]. Medical indications for parentally and orally administered anticoagulants are summarized in Table 5 and Table 6, respectively. 

## 5. Drug–Drug Interactions

Oral anticoagulants come with a lower burden for VTE patients, since no medical personnel or injections are required for their admission. With their rapid mode of action, especially the novel VTE drugs, DOACs are preferred over other currently used drugs. However, compared to parenteral anticoagulants, oral anticoagulants are often accompanied by the risk of undesired drug–drug interaction (DDI) [85]. DDIs are secondary to the specific effect of a therapeutic and can occur when two administered drugs share the same cell membrane transporter or metabolic pathway. Oral anticoagulants need to pass from the gastrointestinal system into the circulation, and, subsequently, they are metabolized by renal or hepatic clearance. Their transport to and from the circulation is mediated by the transporter glycoproteins, such as P-GP [86]. Drugs administered to treat atrial fibrillation may interfere with DOAC transport by P-GP, and this leads to an increase in the anticoagulant activity. For instance, dabigatran activity could increase by up to 70% in patients that are treated with dronedarone. For this reason, to treat atrial fibrillation, the drug amiodarone, a moderate P-GP competitor, is preferred to dronedarone in patients using DOACs [85]. In cancer patients, imatinib and crizotinib, two tyrosine kinases, are contraindicated for DOAC patients, due to their strong inhibitory capacity of P-GP [87]. Enzalutamide, a prostate cancer hormonotherapy, promotes P-GP function, and this could reduce the concentration of the administered DOAC, resulting in an increased risk for recurrent VTE [88]. P450 cytochromes CYP2C9, CYP1A2, and CYP3A4 mediate the elimination of oral anticoagulants, and DOAC or VKA patients that receive P450-cytochromes-inhibiting drugs have an increased bleeding risk [85]. Drugs to treat atrial fibrillation and cancer are most often cited for their DDI with oral anticoagulants, but DDIs with statins and antibiotic, antidepressant, and antiviral drugs have also been described [89,90,91]. 

Certain intrinsic characteristics of individual patients may cause anticoagulants to have a different metabolism. It has been demonstrated that polymorphisms on transporters or P450 cytochromes can lead to unexpected variability in drug efficacy [92]. The same holds true for polymorphisms in the VKCOR1 gene, the target of VKAs, which can cause individuals to have a delayed response towards the drug. Patients with liver or kidney failure demonstrate a delay in the elimination of anticoagulants, thereby increasing the plasma concentration and the risk of bleeding [93,94]. 

## 6. Introducing DOACs in Complex Patient Populations

In many clinical trials to investigate the effects of a certain drug, children are excluded due to their physical vulnerability [95]. For this specific patient population, the use of DOACs to prevent or cure (recurrent) VTE will most likely significantly improve the quality of care, since the drug is administered orally and it does not require intense monitoring. The EINSTEIN Junior study was the first study designed to test DOACs in pediatric VTE patients [96]. Within this study, the efficacy and the safety of rivaroxaban was compared to regular treatment, i.e., treatment with VKA, LMWH, or UFH. Within four years, more than 500 children were recruited, and patients treated with rivaroxaban demonstrated a similar recurrence risk of VTE and bleeding risk, compared to other anticoagulants. Currently, the EINSTEIN Junior study is the largest study to be carried out in children testing DOACs, and, although DOACs appear safe and effective in pediatric VTE patients, we must remain vigilant for their use in this vulnerable population.

Heparins are the only anticoagulants that are authorized during pregnancy, since VKAs and DOACs pass the placental barrier. However, unlike for VKAs, the exact effects of DOACs on the developing fetus are unknown. In 2020, a registry of women who had been pregnant while receiving DOACs was published by Beyer-Westendorf et al. [97]. In that registry, from 2007 to 2020, 336 mothers were included that were exposed to DOACs during pregnancy. In total, 6% of the women displayed a non-normal fetus growth (95% confidence interval 4%–9%, comparable to the normal population). Despite these promising results, physicians remain reluctant to prescribe DOACs in pregnant women. In addition, DOACs are assumed to pass into breast milk, so, for this reason, the drugs are currently not prescribed in nursing mothers [98,99].

The RAPS and TRAPS clinical studies were designed to evaluate DOACs in preventing VTE in triple-positive antiphospholipid syndrome (APS) patients [100,101]. Although the first study provided promising results, making use of a biologic test as a clinical surrogate marker [100], the second study was prematurely stopped due to a major safety issue [101]. It appeared that the use of the anti-FXa DOAC rivaroxaban was associated with an increased risk of thrombotic events, compared to VKA. Based on these results, the use of DOACs is, currently, strongly discouraged in APS patients. However, APS is a complex disease, for which the pathophysiology is not completely understood, so a dedicated study investigating the putative mechanism by which DOACs increase VTE risk in APS patients is required [102]. However, the aforementioned study only tested rivaroxaban and no other DOAC and did not include single- or double-positive APS patients, while real-life data of non-triple-positive APS patients who remained on DOACs are reassuring [101]. These shortcomings argue for new studies on DOACs in triple-positive APS patients. 

In 2013, Eikelboom at al. published a study comparing dabigatran to warfarin in patients with mechanical heart valves [103]. The study was stopped prematurely because, for unknown reasons, an excess of thrombotic events in the dabigatran group was observed (32% vs. 6% in the warfarin group). Interestingly, in a recently published study, rivaroxaban-treated patients with organic heart valve disease-associated atrial fibrillation showed an increase in cardiovascular events and death in DOAC patients, compared to patients under VKA treatment [104]. These data strongly imply that DOACs should be avoided in patients with mechanical or organic heart valves. 

After a pulmonary embolism event, optimal follow-up comprises monitoring for long-term complications, such as chronic thromboembolic pulmonary hypertension (CTEPH) [105]. The efficacy and safety of DOACs to prevent recurrent VTE is challenged because of a higher incidence of CTEPH, compared to VKA-treated patients, possibly because of clinically relevant drug–drug interaction [106,107]. Hence, updated guidelines recommend to prescribe VKA over DOACs in patients with CTEPH [108].

In atherosclerosis, the COMPASS trial (Cardiovascular Outcomes for People Using Anticoagulation Strategies) showed that combining regular aspirin treatment with rivaroxaban (2.5 mg twice daily) was beneficial for coronary, cerebrovascular, and peripheral end points in patients with and without diabetes mellitus [109]. The beneficial effect of aspirin and rivaroxaban has not been confirmed yet for VTE patients. Combination therapy using anticoagulants and anti-aggregants in DVT was assessed by a Cochrane meta-analysis [110]. Following the initial standard treatment with anticoagulants, there appeared to be low-certainty evidence that antiplatelet agents, in addition to standard anticoagulation and other clinical practices (like stocking socks), reduce recurrent VTE, with no clear differences of adverse events such as major bleeding or recurrent VTE. For PE, no recent studies have been published on the subject of combination therapy. Fear of bleeding events should not be an obstacle to the production of reassuring data on the risk of bleeding associated with anticoagulant and anti-aggregant combination therapy [111]. However, dedicated studies are needed to explore the potential of combining drugs to prevent or cure VTE.

## 7. A New Generation of Anticoagulants

When introducing a new drug on the market, it should be beneficial compared to existing agents. This benefit can consist of improvements in, for instance, higher efficacy, easier admission, less strict monitoring, and reduced side effects. In VTE patients, bleeding as a consequence of anticoagulant treatment is a major issue, so, for this reason, the scientific community of thrombosis and hemostasis has proposed multiple innovative strategies, with the goal to combat thrombosis in humans without (excessive) bleeding as a side effect. These novel strategies can be based on new techniques, such as making use of antisense oligonucleotides, antibodies, or aptamers, or based on targeting other components from the coagulation cascade, such as the FVIIa–tissue factor complex, FIXa, FXIIa, or FXIa (Figure 1) [4]. The latter coagulation factor is of especially great interest, since inhibition of FXIa seems to prevent thrombosis without (excessive) bleeding as a side effect [112]. Abelacimab is an anti-FXIa antibody that is currently being tested in Phase III studies in patients with cancer-associated thrombosis (https://clinicaltrials.gov/ct2/show/NCT05171049 and https://clinicaltrials.gov/ct2/show/NCT05171075 (both accessed on 1 September 2022)). Cancer-associated thrombosis (CAT) patients form a frail population that is challenging to treat, since they have an increased risk for both (recurrent) VTE and anticoagulation-associated bleeding. According to current guidelines, to limit the risk of excessive bleedings, CAT patients should be treated with great caution when using DOACs or LMWHs [113,114,115,116]. FXIa drugs are associated with a lower bleeding risk and may be administered at a higher dosage in CAT patients to prevent recurrent VTE, thereby improving patient care. If these clinical studies on CAT patients prove to be successful, FXIa drugs will most likely be tested in other frail VTE patient populations [112]. 

## 8. Conclusions

In the last 100 years, several anticoagulant drugs have been introduced on the market. All drugs have a different history, mechanism, admission route, pharmacokinetics, and medical indication. Anticoagulants have aided considerably in combatting thrombotic diseases, although they all have their specific flaws, so, for this reason, it is of great interest to the scientific community of thrombosis and hemostasis to investigate novel anticoagulants. The perfect anticoagulant should be well-tolerated by the entire patient population, be orally administered with a wide therapeutic window, and have favorable absorption, distribution, and elimination. In addition, efficacy of the drug should be easy to test, a proper neutralizing agent should be available, and, perhaps most importantly, the drug should have limited to no side effects such as bleeding. New anticoagulants such as the FXIa inhibitors might present an improvement compared to the current treatment options. 

## Figures and Tables

**Figure 1 cells-11-03214-f001:**
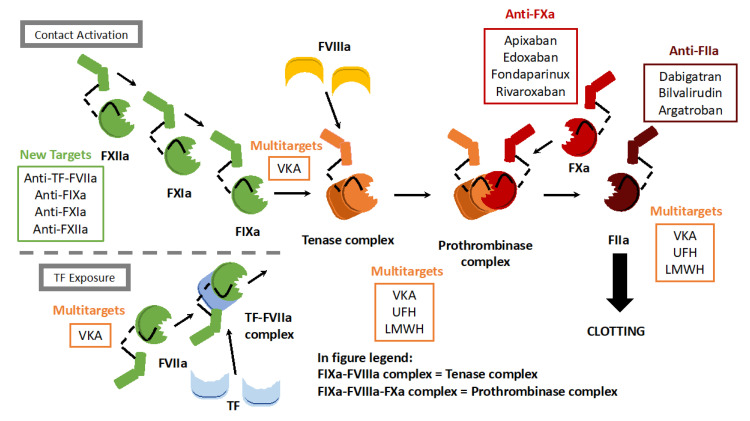
Anticoagulants and their targets within the coagulation cascade. The coagulation cascade consists of three parts: the tissue factor (TF) pathway, the contact activation pathway, and the common pathway. Currently used anticoagulants target primarily coagulation factors from the common pathway: FXa (apixaban, edoxaban, fondaparinux, and rivaroxaban) and thrombin (FIIa; dabigatran, bilvalirudin, and argatroban). UFH, VKA, and LMWH have multiple targets within the coagulation cascade. Anticoagulants that are currently under development or undergoing clinical trials target factors from the TF pathway (TF-FVIIa complex) or the contact activation pathway (FIXa, FXIa, and FXIIa). Tenase complex: FVIIIa-FIXa complex. Prothrombinase complex: FVIIIa-FIXa-FXa complex.

**Table 1 cells-11-03214-t001:** Pharmacokinetics of UFH, enoxaparin, and fondaparinux. Other LMWHs have similar pharmacological characteristics as enoxaparin, the most frequently used. UFH: unfractionated heparin, Vd: volume of distribution, IV: intravenous, SC: subcutaneous.

	UFH	Enoxaparin	Fondaparinux
Type	Small molecule	Small molecule	Small molecule
Origin	Obtained from liver, lung, mast cells, and other cells of vertebrates	Obtained from liver, lung, mast cells, and other cells of vertebrates	Synthetic molecule
FDA approval	N/A, first used in clinical practice in 1941	1993	2001
Target	Antithrombin	Antithrombin	Antithrombin
Available administration route	IV/SC	(IV)/SC	(IV)/SC
Absorption	N/A	N/A	N/A
Bioavailability	Unpredictable	90%–92%	100%
Vd	40–70 mL/min	4.3 L	7–11 L
Protein binding	>90%	<UFH	>94% (specifically to antithrombin)
Time to peak activity (h)	Rapid after bolus, 4–6 after infusion	3–5	2–3
Monitoring	aXa UFH	aXa LMWH	aXa Fondaparinux
Half-life (h)	1.0–1.5	5	17–21
Elimination	Reticuloendothelial system + renal	Renal	Renal
Neutralizing agent	No hemodialysis Sulfate protamine	Hemodialysis Sulfate protamine	Hemodialysis No sulfate protamine

**Table 2 cells-11-03214-t002:** Pharmacokinetics of warfarin. Vd: volume of distribution, PO: per os (orally), INR: international normalized ratio.

	Warfarin
Type	Racemic mixture
Origin	Synthetic molecule
FDA approval	1954
Target	Vitamin K-dependent coagulation factors II, VII, IX, and X
Available administration route	PO
Absorption	Stomach and proximal small bowel
Bioavailability	80%–100%
Vd	8–10 L
Protein binding	95%–97%
Time to peak concentration (h)	1–2
Monitoring	INR test
Half-life (h)	20–60
Elimination	Major hepatic metabolism, biotransformation > 90% of inactive metabolites (CYP 2C9)
Neutralizing agent	Vitamin K

**Table 3 cells-11-03214-t003:** Pharmacokinetics of bivalirudin and argatroban. Vd: Volume of distribution, IV: intravenous, aPTT: activated partial thromboplastin time, ACT: activated clotting time.

	Bivalirudin	Argatroban
Type	Small molecule	Small molecule
Origin	Synthetic molecule	Synthetic molecule
FDA approval	2000	2000
Target	Thrombin	Thrombin
Available administration route	IV	IV
Absorption	N/A	N/A
Bioavailability	40%–80%	100%
Vd	0.24 L/Kg	0.174 L/Kg
Protein binding	No plasma proteins (just thrombin)	55% (20% albumin, 35% alpha-acid glycoprotein)
Time to peak activity	2–4 min	3–4 h
Monitoring	aPTT, ACT, ecarin clotting time	aPTT, ACT
Half-life (hours)	1.0–1.5	3–5
Elimination	Renal and proteolytic cleavage	70% hepatic hydroxylation, 16% renal excretion as unchanged drug, 14% biliary excretion as unchanged drug
Neutralizing agent	N/A	N/A

**Table 4 cells-11-03214-t004:** Pharmacokinetics of dabigatran, rivaroxaban, apixaban, and edoxaban. Vd: volume of distribution, PO: per os (orally). PCC: prothrombin complex concentrates.

	Dabigatran	Rivaroxaban	Apixaban	Edoxaban
Type	Small molecule	Small molecule	Small molecule	Small molecule
Origin	Synthetic molecule	Synthetic molecule	Synthetic molecule	Synthetic molecule
FDA approval	2010	2011	2014	2014
Target	FIIa	FXa	FXa	FXa
Available administration route	PO	PO	PO	PO
Absorption	Lower stomach + duodenum	Proximal small bowel + gastric	Proximal small bowel + gastric	Proximal small bowel
Bioavailability	6.5%	66%	50%	60%
Vd	50–70 L	50 L	21 L	>107 L
Protein binding	35%	>90%	87%	40%–60%
Time to peak concentration (h)	1–2	2–4	1–4	1–2
Monitoring	N/A	N/A	N/A	N/A
Half-life (h)	12–17	5–9	8–15	10–14
Elimination	Up to 20% glucuronidation and then biliary excretion 80% unchanged renal excretion	66% undergoes metabolic degradation (30% CYP CYP3A4/5, metabolism) 66% renal excretion 28% feces	30%–35% O-demethylation 15% CYP CYP3A4/5, metabolism 1/3 biliary excretion 1/3 renal excretion	Conjugation, oxidation by CYP 3A4 and hydrolysis 1/2 biliary excretion 1/2 renal excretion
Neutralizing agent	Idarucizumab	Andexanet alfa, PCC	Andexanet alfa, PCC	Andexanet alfa (?), PCC

**Table 5 cells-11-03214-t005:** Indications for the parenterally administered anticoagulants unfractionated heparin (UFH), enoxaparin, fondaparinux, argatroban, and bilvarudin. * based on study using another LMWH, dalteparin.

Indication	UFH	Enoxaparin	Fondaparinux	Bilvarudin	Argatroban
Arterial thrombosis					
Prophylaxis of ischemic complications in the setting of unstable angina or NSTEMI	yes	yes	/	/	/
Treatment of atrial fibrillation with embolization	yes	/	/	/	/
Prophylaxis of peripheral arterial embolism	yes	/	/	/	/
Prophylaxis of clotting in cardiac surgery	yes	/	/	/	/
Prophylaxis or treatment of thrombosis in adult patients with HIT, including during PCI procedures	/	/	/	yes	yes
Venous thrombosis					
Extended treatment of symptomatic venous thromboembolism to reduce recurrence in patients	yes	yes	yes	/	/
Prophylaxis of postoperative VTE in patients undergoing abdominal surgery, hip replacement surgery, or knee replacement surgery	yes	yes	yes	/	/
Prophylaxis of VTE in acutely ill medical patients with severely restricted mobility	/	yes	/	/	/
Extended treatment of symptomatic venous thromboembolism to reduce recurrence in patients with cancer	/	yes *	/	/	/
Other indications					
Extracorporeal circulation and dialysis procedures	yes	/	/	/	/
Treatment of acute and chronic consumptive coagulopathies	yes	/	/	/	/

**Table 6 cells-11-03214-t006:** Indications for the orally administered anticoagulants warfarin, dabigatran, rivaroxaban, apixaban, and edoxaban.

Indication	Warfarin	Dabigatran	Rivaroxaban	Apixaban	Edoxaban
Arterial thrombosis					
Reduced risk of stroke and systemic embolism in patients with NVAF	yes	yes	yes	yes	yes
Prophylaxis and treatment of embolic complications associated with atrial fibrillation or cardiac valve replacement	yes	/	/	/	/
Reduced risk of death, recurrent myocardial infarction, and stroke or systemic embolism after myocardial infarction	yes	/	/	/	/
Venous thrombosis					
Prophylaxis of DVT and PE in patients who have undergone hip replacement surgery	yes	yes	yes	yes	/
Treatment of DVT and PE	/	yes	yes	yes	yes
Reduced risk of recurrent DVT and/or PE in patients at continued risk for recurrent of VTE	yes	yes	yes	yes	yes
Reduced risk of recurrent DVT and/or PE in patients at continued risk for recurrent of VTE in cancer patients	/	/	yes	yes	yes

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
