# Peer review of "Anticoagulants: A Short History, Their Mechanism of Action, Pharmacology, and Indications"

_cells, 2022, doi:10.3390/cells11203214_

Round 1
Reviewer 1 Report
Comment to respected authors:
This is a well-written and comprehensive review that has introduced the current anticoagulants in use while clearly discussing different mechanistic approaches, highlighting their advantages and disadvantages or in-act limitations of all drugs with a fair justification regarding the limitation of currently introduced DOACs. However, it should be kept in mind that the current debate is not just the absolute use of DOACs versus previous standard methods such as LMWH infusion, and there are many studies that have used the combination of lower-dose DOAC treatments with conventional methods during the designated period of treatment for patients, the benefits of which can be mentioned in reducing costs, increasing the ease and efficiency of anticoagulation treatments. This is an important point that has been neglected, since the issues with cost effectiveness of DOACs limits their applications, especially in developing countries where authorities tend to stick to conventional methods without sufficient knowledge about such flexibility in prescribing DOACs. It is much appreciated if the authors can provide a specific section for the feasibility of combined therapies based on available protocols and literature.
Author Response
This is a well-written and comprehensive review that has introduced the current anticoagulants in use while clearly discussing different mechanistic approaches, highlighting their advantages and disadvantages or in-act limitations of all drugs with a fair justification regarding the limitation of currently introduced DOACs.
The authors would like to thank the reviewer for the positive review of our manuscript.
However, it should be kept in mind that the current debate is not just the absolute use of DOACs versus previous standard methods such as LMWH infusion, and there are many studies that have used the combination of lower-dose DOAC treatments with conventional methods during the designated period of treatment for patients, the benefits of which can be mentioned in reducing costs, increasing the ease and efficiency of anticoagulation treatments. This is an important point that has been neglected, since the issues with cost effectiveness of DOACs limits their applications, especially in developing countries where authorities tend to stick to conventional methods without sufficient knowledge about such flexibility in prescribing DOACs. It is much appreciated if the authors can provide a specific section for the feasibility of combined therapies based on available protocols and literature.
We agree with the reviewer that we did not include studies on the feasibility of combined therapies, and we agree that adding a section discussing this topic would be informative for the readers of Cells. For this reason, we added a the following paragraph to chapter 7:
In atherosclerosis, the COMPASS trial (Cardiovascular Outcomes for People Using Anticoagulation Strategies) showed that combining regular aspirin treatment with rivaroxaban (2.5 mg twice daily) was beneficial for coronary, cerebrovascular, and peripheral end points in patients with and without diabetes mellitus (DOI: 10.1161/CIRCULATIONAHA.120.046448). The beneficial effect of aspirin and rivaroxaban has not been confirmed yet for VTE patients. Combination therapy using anticoagulants and anti-aggregant in DVT was assessed by a Cochrane meta-analysis (DOI: 10.1002/14651858.CD012369.pub2). Following the initial standard treatment with anticoagulants, there appeared to be a low‐certainty evidence that antiplatelet agents in addition to standard anticoagulation and other clinical practices (like stocking socks) reduces recurrent VTE, with no clear differences of adverse events such as major bleeding or recurrent VTE. For PE, no recent studies have been published on the subject of combination therapy. Fear of bleeding events should not be an obstacle to the production of reassuring data on the risk of bleeding associated with anticoagulant and anti-aggregant combination therapy (DOI: 10.1016/j.thromres.2021.11.001). However, dedicated studies are needed to explore the potential of combining drugs to prevent or cure VTE.
Reviewer 2 Report
Very good and comprehensive review on pharmacokinetic and indications of anticoagulants with history of these drugs. Full bibliography.
Critical remarks:
1. Not full list of key words
2. Page 6 – row 210 “Warfarin is the only VKA that is currently recommended to use in the clinic.” This is not true. In many country as in Bulgaria warfarin is not available, we use acenocumarol, in Germany phenprocoumon is used. The author should point out the particular country where Warfarin is the only VKA that is currently recommended to use or to expand this section with all VKA in the text and in the table 2.
3. The author used term “antidote” – in my opinion “neutralizing agent” is the appropriate term.
4. In table 4 – “Pharmacokinetics of Dabigatran, Rivaroxaban, Apixaban, Edoxaban, and Betrixaban” - the column for Betrixaban is missing.
5. In table 6 – “Table 6. Indications for the orally administered anticoagulants Warfarin, Dabigatran, Riveroxaban, Apixaban, Edoxaban, and Betrixaban” – there is a spelling mistake in Rivaroxaban.
Author Response
Very good and comprehensive review on pharmacokinetic and indications of anticoagulants with history of these drugs. Full bibliography.
The authors would like to thank the reviewer for the positive review of our manuscript, and for the critical remarks that allow us to submit an improved version of our manuscript. Below, we provide a “point-by-point” response to the critical remarks.
Critical remarks:
- Not full list of key words
We updated and extended our selection for key words: “Venous thromboembolism, Anticoagulant drugs, heparin, vitamin K-antagonists, DOACs, Pharmacology, Drug-drug interactions, novel anticoagulants”.
- Page 6 – row 210 “Warfarin is the only VKA that is currently recommended to use in the clinic.” This is not true. In many country as in Bulgaria warfarin is not available, we use acenocumarol, in Germany phenprocoumon is used. The author should point out the particular country where Warfarin is the only VKA that is currently recommended to use or to expand this section with all VKA in the text and in the table 2.
We agree with the reviewer that our statement is incorrect in its current form. For this reason, we deleted sentence 210.
Throughout the rest of the manuscript, we altered the text to emphasize that we are discussing VKAs and not only warfarin. Importantly, these minor changes did not alter the content of the manuscript.
- Line 141: “warfarin” is replaced with “VKAs”
- Line 200: “warfarin” is replaced with “VKAs”
- Line 212: “warfarin” is replaced with “VKAs”
- Line 219: “warfarin” is replaced with “VKAs”
- Line 306: “warfarin” is replaced with “VKAs”
- Line 324: “warfarin” is replaced with “VKAs”
For tables 2 and 6 we did not make changes. Alternatively, in lines 220/221 we pointed out that warfarin is the first described and most commonly used VKA worldwide and that will serve as a reference other VKAs in our table with the pharmacological properties (table 2) and indications for orally admitted anticoagulants (table 6).
- The author used term “antidote” – in my opinion “neutralizing agent” is the appropriate term.
According to the reviewer’s suggestion, we altered the term antidote in neutralizing agent.
- In table 4 – “Pharmacokinetics of Dabigatran, Rivaroxaban, Apixaban, Edoxaban, and Betrixaban” - the column for Betrixaban is missing.
Since Betrixaban is not discussed in the text, we decided to delete Betrixaban from the legends of tables 4 and 6.
- In table 6 – “Table 6. Indications for the orally administered anticoagulants Warfarin, Dabigatran, Riveroxaban, Apixaban, Edoxaban, and Betrixaban” – there is a spelling mistake in Rivaroxaban.
We corrected the spelling of Rivaroxaban.